 SHORT REPORT

# Pharmacological targeting of the transcription factor SOX18 delays breast cancer in mice

Jeroen Overman[1], Frank Fontaine[1†], Mehdi Moustaqil[1,2†], Deepak Mittal[3], Emma Sierecki[1,2], Natalia Sacilotto[4], Johannes Zuegg[1], Avril AB Robertson[1], Kelly Holmes[5], Angela A Salim[1], Sreeman Mamidyala[1], Mark S Butler[1], Ashley S Robinson[6], Emmanuelle Lesieur[1], Wayne Johnston[1], Kirill Alexandrov[1], Brian L Black[6], Benjamin M Hogan[1], Sarah De Val[4], Robert J Capon[1], Jason S Carroll[5], Timothy L Bailey[1], Peter Koopman[1], Ralf Jauch[7,8], Matthew A Cooper[1], Yann Gambin[1,2], Mathias Francois[1*]

[1]Institute for Molecular Bioscience, The University of Queensland, Brisbane, Australia; [2]Single Molecule Science, Lowy Cancer Research Centre, The University of New South Wales, Sydney, Australia; [3]Immunology in Cancer and Infection Laboratory, QIMR Berghofer Medical Research Institute, Herston, Australia; [4]Ludwig Institute for Cancer Research, Nuffield Department of Clinical Medicine, The University of Oxford, Oxford, United Kingdom; [5]Cancer Research UK, The University of Cambridge, Li Ka Shing Centre, Cambridge, United Kingdom; [6]Cardiovascular Research Institute, The University of California, San Francisco, San Francisco, United States; [7]Genome Regulation Laboratory, Drug Discovery Pipeline, CAS Key Laboratory of Regenerative Biology, Joint School of Life Sciences, Guangzhou Institutes of Biomedicine and Health, Chinese Academy of Sciences, Guangzhou, China; [8]Guangzhou Medical University, Guangzhou, China

**\*For correspondence:**
m.francois@imb.uq.edu.au

[†]These authors contributed equally to this work

**Competing interest:** The authors declare that no competing interests exist.

**Abstract** Pharmacological targeting of transcription factors holds great promise for the development of new therapeutics, but strategies based on blockade of DNA binding, nuclear shuttling, or individual protein partner recruitment have yielded limited success to date. Transcription factors typically engage in complex interaction networks, likely masking the effects of specifically inhibiting single protein-protein interactions. Here, we used a combination of genomic, proteomic and biophysical methods to discover a suite of protein-protein interactions involving the SOX18 transcription factor, a known regulator of vascular development and disease. We describe a small-molecule that is able to disrupt a discrete subset of SOX18-dependent interactions. This compound selectively suppressed SOX18 transcriptional outputs in vitro and interfered with vascular development in zebrafish larvae. In a mouse pre-clinical model of breast cancer, treatment with this inhibitor significantly improved survival by reducing tumour vascular density and metastatic spread. Our studies validate an interactome-based molecular strategy to interfere with transcription factor activity, for the development of novel disease therapeutics.

## Introduction

The SOXF group (SOX7, —17 and —18) of transcription factors (TFs) are key regulators of endothelial cell differentiation during development (*François et al., 2008*; *Corada et al., 2013*; *Hosking et al., 2009*; *Matsui et al., 2006*; *Cermenati et al., 2008*; *Herpers et al., 2008*), and are thus critical for

the formation of vasculature. Mutation or deletion of *SoxF* genes compromises arteriovenous specification, blood vascular integrity and lymphangiogenesis, and inhibits tumour growth and metastasis in animal models of cancer (*Duong et al., 2012*; *Yang et al., 2013*; *Zhang et al., 2009*; *Young et al., 2006*). More recently, high levels of SOX18 have been associated with poor prognosis for cancer in human patients (*Eom et al., 2012*; *Pula et al., 2013*; *Jethon et al., 2015*). Pharmacological inhibition of SOX18 protein function therefore presents a potential avenue for management of the vascular response in cancer.

Transcription factors often operate in mutually redundant families, thwarting conventional approaches to developing transcription factor-based therapies. Any attempt to develop pharmaceutically useful SOX18 inhibitors must overcome two obstacles — first, that SOX18 loss of function is compensated by the action of the remaining SOXF (*Hosking et al., 2009*), and second, that each SOXF factor is likely to have several partners that may themselves act redundantly. To address these challenges, we sought to develop a means of broad-scale functional inhibition of SOX18 transcription factor through the simultaneous interference with multiple SOX18 protein-protein interactions (PPIs).

SOX proteins activate individual target genes by recruiting specific interacting partners (*Sarkar and Hochedlinger, 2013*), but only two protein-protein interactions for the SOXF group (SOX18-MEF2C and SOX17-OCT4) have been identified to date (*Hosking et al., 2001*; *Jauch et al., 2011*). We first mapped the SOX18 interactome (the network of SOX18 interacting partners), using a combination of unbiased proteomic technologies. Chromatin immunoprecipitation coupled to mass spectrometry (ChIP-MS) provided a first-pass screen for proteins associated with chromatin- bound SOX18 in human umbilical vein endothelial cells (HUVECs) (*Mohammed et al., 2013*), then, ALPHA-Screen resolved SOX18-dependent complexes into pairwise interactions using in vitro translated full-length proteins (*Figure 1A*; *Mureev et al., 2009*; *Kovtun et al., 2011*; *Sierecki et al., 2013*; *Sierecki et al., 2014*; *Gambin et al., 2013*). ChIP-MS analysis revealed 289 proteins, representing a variety of gene ontology (GO) classes of molecular function, that associate directly or indirectly with SOX18 (*Figure 1B*, *Figure 1—figure supplement 1A–C*). To increase our chance of identifying direct interactors, we focused on proteins known to be nucleic acid and/or protein binding (*Figure 1B*, purple). From this subset, we chose eight known transcription factors, helicases, co-repressors, RNA binding and DNA-repair molecules (*Figure 1—figure supplement 1A and B*). Using ALPHA-Screen, we observed that SOX18 interacts with itself, and also forms pairwise interactions with DDX1, DDX17, ILF3, STAT1, TRIM28, and XRCC5 (*Figure 1C*, left column '+', *Figure 1—figure supplement 1D*).

In addition, we studied potential pairwise interactions of 6 well-known TFs able to regulate endothelial cell function (ESR1, NR2F2, RBPJ, SOX7, SOX17 and CTNNB1), and the only identified SOX18 protein partner MEF2C (*Hosking et al., 2001*). The well-characterized SOX9 homo-dimer (*Bernard et al., 2003*) was included as a positive control to validate the ALPHA-Screen signal (*Figure 1—figure supplement 1D*). SOX18 was found to interact with all endothelial transcription factors tested, with the possible exception of SOX17 and CTNNB1, which showed a binding affinity below the arbitrary threshold (*Figure 1C*, '-').

Having identified an array of proteins able to interact with SOX18, we then went on to test the activity of a small-molecule compound, **Sm4** (*Figure 1—figure supplement 1E*), on these interactions. **Sm4**, derived from a natural product found in the brown alga Caulocystis cephalornithos, was identified in a high-throughput screen for potential SOX18 blockers (Fontaine et al., to be published in full elsewhere). We found that **Sm4** significantly disrupted 6 out of the 12 validated SOX18 interactions (*Figure 1C*, right column), with $IC_{50}$ values ranging from 3.3 mM for SOX18-SOX18 to 65.9 MM for SOX18-RBPJ dimers (*Figure 1D*, *Figure 1—figure supplement 1F*). To assess a differential effect of **Sm4** on the distinct SOXF members, we explored an additional set of PPIs between all three SOXF proteins and MEF2C, RBPJ and OCT4 (*Figure 1—figure supplement 2*). Like SOX18, SOX7 is able to interact with RBPJ and SOX18 itself, both of which interactions are at least partially disrupted by **Sm4**. We further found that all three SOXF proteins can form a heterodimer with OCT4, whereas only the SOX17-OCT4 interaction is affected by **Sm4**. Importantly, neither SOX7 nor SOX17 have the capacity to form a homodimer, and thus this component of **Sm4** mode of action is highly specific to SOX18-SOX18 interaction. Further corroborating this, SOX9 homodimerization was unperturbed by **Sm4** at up to 200 mM (*Figure 1C and D*, *Figure 1—figure supplement 1D*). These results show that **Sm4** selectivity leans towards a subset of SOX18-associated PPIs, but has the capability to interfere with SOX7 or SOX17 protein partner recruitment. This feature of

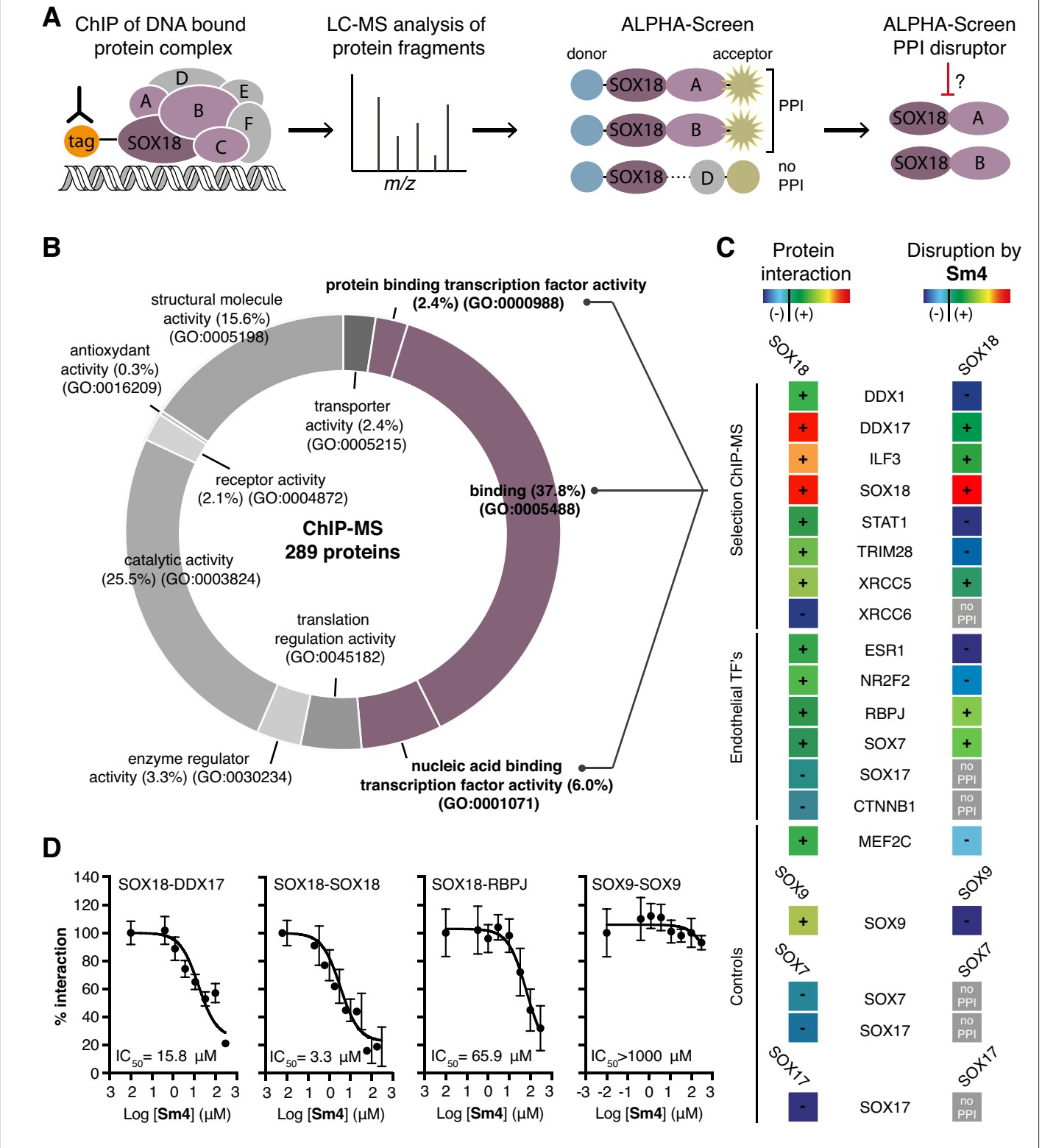

**Figure 1.** Mapping of SOX18 interactome and disruption of interactions by **Sm4**. (**A**) Schematic of the experimental strategy to deconvolute SOX18-dependent protein-protein interactions (PPIs) combining Chromatin immunoprecipitation-mass spectrometry (ChIP-MS) and Amplified Luminescent Proximity Homogeneous Assay (ALPHA-Screen) methods. (**B**) GO-term analysis for molecular function on the 289 proteins identified by SOX18-cMyc ChIP-MS in human umbilical vein endothelial cells (HUVECs). Non-specific interactors found in Myc-tag only transfected cells were subtracted. Proteins with nucleic acid binding or protein binding capacity (purple) were considered for consecutive direct interaction studies to enhance likeness

*Figure 1 continued on next page*

*Figure 1 continued*

of identifying direct interactors. (**C**) Left column: heatmap representation of SOX18 pairwise PPIs as tested by ALPHA-Screen, on a selection of ChIP-MS SOX18 associated proteins, endothelial transcription factors and positive/negative control proteins. Right column: heatmap representation of **Sm4** activity on SOX18-dependent protein-protein interactions, as tested at 100 pM. Interaction and disruption threshold is indicated in the scale bar by a black line. Levels of interaction and disruption above the threshold are demarked by '+', and below the threshold by '—'. Tagged proteins were expressed in the *Leishmania tarentolae* cell-free protein expression system. (**D**) Representative ALPHA-Screen concentration-response curve for SOX18 PPI disruption by **Sm4**. Data shown are mean ± s.e.m.

The online version of this article includes the following figure supplement(s) for figure 1:

**Figure supplement 1.** QC of SOX18 PPIs and effect of **Sm4**.

**Figure supplement 2.** Differential disruption of SOXF PPI by **Sm4**.

**Sm4** is potentially advantageous in preventing SOXF redundancy mechanism (*Hosking et al., 2009*; *Kim et al., 2016*).

To assess how SOX18 PPI disruption translates into transcriptional dysregulation, we next performed a combination of genome-wide RNA-seq and ChIP-seq analyses in HUVECs. The most common binding motif identified from the SOX18 ChIP-seq peaks corresponds to the previously reported SOX motif 5'-AACAAT-3' (*Figure 2—figure supplement 1A*) and the validity of this ChIP- seq dataset was further confirmed by GO term analysis and identification of known SOX18 target genes such as *Proxl* and *Vcaml* (*Supplementary file 1a*, *Figure 2—figure supplement 1B*; *François et al., 2008*; *Hosking et al., 2004*). We compared the global transcriptional effect of **Sm4** treatment to DMSO control in SOX18 overexpressing cells (*Figure 2—figure supplement 1C–E*, *Supplementary file 1b*), and over-laid this list of differentially expressed genes with the SOX18 ChIP- seq dataset. Using this overlay, we calculated the distance between the transcription start site (TSS) of a gene and a TF binding event, as a proxy for the likelihood of direct transcriptional regulation. To be able to analyse how this distance is altered by **Sm4**, we established a reference distance between the TSS of a random gene set and SOX18 binding events (*Figure 2A*). In parallel, we performed the same analysis for SOX7 (generated in-house), and for all seven transcriptional regulators available from the ENCODE consortium (cMYC, GATA2, c-FOS, c-JUN, CTCF, EZH2, MAX, c-MYC). This allowed us to distinguish between transcriptional targeting of SOX18 and potential off target effects on other endothelial specific transcription factors.

The cumulative SOX18 peak-to-TSS distance demonstrated that, overall, SOX18 peaks are 3.6 fold closer (p-value <0.001) to the TSS of **Sm4** down-regulated genes than to randomly distributed TSSs (*Figure 2B*, top left). These results are an indirect indication that the **Sm4** affected genes are dysregulated through a specific effect on SOX18 transcriptional activity. This correlation was not observed for 7 of the other transcription factors tested (*Figure 2B*, *Figure 2—figure supplement 1F*, *Supplementary file 1c*), signifying that **Sm4** does not have an off-target effect on these TFs activity. Interestingly, the TSS of **Sm4** down-regulated genes were 2.05 fold closer to c-JUN binding events (p-value = 0.011, *Supplementary file 1c*). Although only mildly significant, this could suggest possible co-regulation by SOX18 and c-JUN on this subset of **Sm4** down-regulated genes. Indeed, analysis of known motifs in SOX18 ChIP-seq peaks revealed an over-representation of c-JUN binding motifs (3.23% of SOX18 peaks, p-value = 1e-302) and ALPHA-Screen analysis further established that SOX18 and c-JUN could physically interact (*Figure 2—figure supplements 1 and 2*). We found that the expression levels of the other TFs tested were unaltered by **Sm4** treatment (*Supplementary file 1c*). This is an important observation because it demonstrates that there was no bias introduced by an off-target modulation of the transcript levels for these transcription factors in presence of **Sm4**.

To address the issue of potential transcriptional off-target effects of **Sm4** on SOX TF family members we focused on closely related SOXF and SOXE proteins. **Sm4** did not affect the transcriptional activity of either SOX17 or SOX9 proteins at any tested concentration (<50 p,M) in cell-based reporter assays (*Figure 2—figure supplement 3*; *Robinson et al., 2014*; *Lefebvre et al., 1997*). Together, these results provide strong evidence that **Sm4** selectively targets SOX18-mediated transcription over other key endothelial transcription factors and SOX proteins.

To investigate whether **Sm4** is also able to perturb Sox18 transcriptional activation in vivo, we used the *tg(−6.5kdrl:eGFP)* transgenic zebrafish reporter line, previously validated as a readout for the combined activity of Sox7 and Sox18 (*Duong et al., 2014*). We treated these larvae at 20 hr post fertilization (hpf) and observed that **Sm4** treatment significantly reduced SOX18-dependent *egfp*

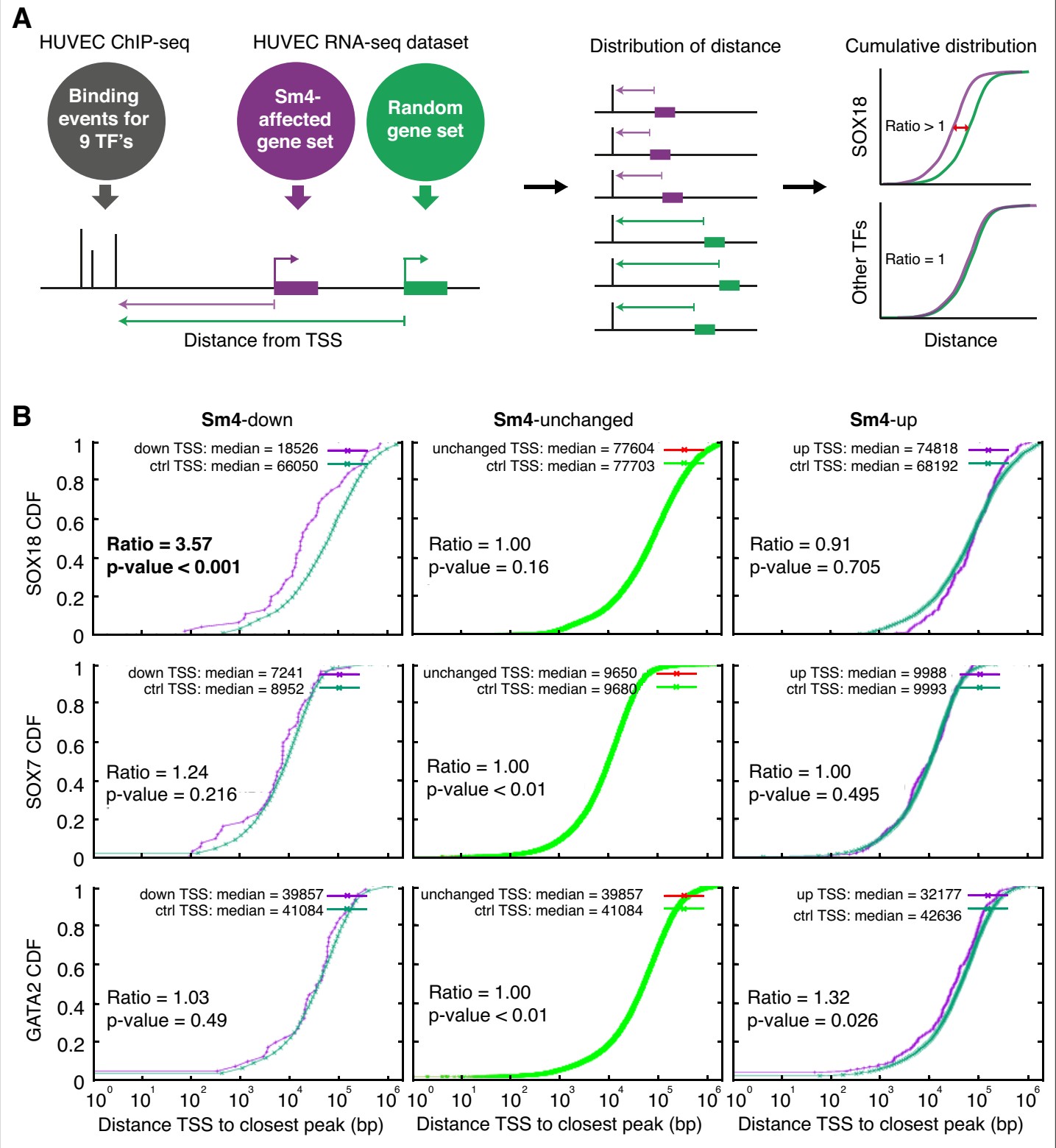

**Figure 2.** Sm4 selectively affects SOX18 transcriptional output in vitro. (**A**) Schematic representation of the correlation analysis between genome-wide TF ChIP-seq data and **Sm4** affected genes from transcriptomics data. The chromatin around the transcription start sites (TSS) of **Sm4** affected genes (purple) was investigated for transcription factor binding peaks (grey), to calculate the 'distance from TSS' to closest binding site for a given transcription factor. This distance from TSS was used as a proxy for the likelihood of transcriptional regulation, and thus make an association between **Sm4** affected genes and transcription factors. Included in the analysis where the ChIP-seq peaks of SOX18 and SOX7, and of all transcription factors available from

*Figure 2 continued on next page*

*Figure 2 continued*

the Encode consortium (GATA2, c-FOS, c-JUN, CTCF, EZH2, MAX and c-MYC), performed in HUVECs. A random group of genes was analysed as a control distribution as would be found by chance. (**B**) **Sm4** affected genes were grouped into down-regulated (**Sm4**-down), unaffected (**Sm4**-unchanged) and up-regulated (**Sm4**-up). The plots show the cumulative distribution of the distance between the TSS of **Sm4** affected genes (purple line, absolute fold change >2) and the closest genomic location of binding sites for SOX18, and control transcription factors SOX7 and GATA2. The median distance from the TSS of differentially expressed genes to the nearest binding event of a given transcription factor was compared to the median distance that is expected by chance from a random gene set (green line). **Sm4** down regulated genes are significantly closer (bold) to the SOX18 peaks, but not to SOX7 or GATA2 peaks.

The online version of this article includes the following figure supplement(s) for figure 2:

**Figure supplement 1.** Transcriptome-wide analysis of **Sm4** selectivity in vitro.

**Figure supplement 2.** c-JUN motifs are enriched in SOX18 binding sites.

**Figure supplement 3.** **Sm4** does not interfere with SOX9 or SOX17 activity in vitro.

---

transcript levels (61%), similar to the effects of combined *sox7/18* depletion using morpholino oligo-nucleotides (MO) (*Figure 3A and B*). Importantly, these zebrafish embryos developed normally and we found no evidence of toxicity.

We then used a second transgenic zebrafish reporter line *tg(Dll4in3:eGFP)*, which harbours a regulatory element located in the intron 3 of *dll4* gene. The activity of this *Dll4in3* enhancer does not fully recapitulate the endogenous *dll4* expression (*Wythe et al., 2013*; *Sacilotto et al., 2013*), but it does provide a useful tool to study the combinatorial activity of Sox7, Sox18 and the Notch effector Rbpj. Combined genetic interference with *sox7*, *sox18* and *rbpj* has been shown to abolish *Dll4in3* activation, while single or double MO knockdowns have a much milder effect (*Sacilotto et al., 2013*). This mild repressive effect was recapitulated by treatment with **Sm4** alone (*Figure 3C, D*). In addition, when *rbpj* MO injections at suboptimal dose were combined with **Sm4** treatment, the repressive effect was significantly increased by 11.5% (*Figure 3C and D*). These data show that **Sm4** interferes with Sox7/18 and Rbpjco-ordinated activation of the *Dll4in3* enhancer. As a negative control in vivo, we used the Sox9-dependent *tg(col2a1:yfp)* reporter line, and observed that continuous **Sm4** treatment between 2 and 6 days post fertilization did not perturb the transcriptional activity of Sox9 or the process of chondrogenesis (*Figure 3—figure supplement 1*). Together, this supports the proposed mechanism of action for **Sm4** as a selective SOX18 inhibitor in vivo.

To further demonstrate the small molecule inhibition of Sox18 function in vivo, we next investigated whether **Sm4** treatment would be able to cause a vascular phenotype, similar to that of **sox7 /sox18** genetically disrupted zebrafish (*Hermkens et al., 2015*). This phenotype is characterised by an arteriovenous specification defect, with reduced expression levels of arterial markers (*Cermenati et al., 2008*; *Herpers et al., 2008*; *Pendeville et al., 2008*). We treated zebrafish larvae harbouring the arterial/venous reporter *tg(fli1a:eGFP,—6.5kdrl:mCherry)* with 1.5 pM **Sm4** during the relevant developmental window, starting from 16 hpf (*Figure 3—figure supplement 2A*). These larvae acquired an enlarged posterior cardinal vein (PCV) at the expense of the dorsal aorta (DA) (*Figure 3E–G*, *Figure 3—figure supplement 2B*), with arteriovenous shunts and incomplete trunk circulation (*Figure 3—figure supplement 2C and D*). qRT-PCR analysis of blood vascular markers at 24 and 48 hpf revealed a significant dysregulation of arterial and venous genes in **Sm4**-treated conditions compared to DMSO, particularly *efnb2a*, *hey1* and *efnb4a* (*Figure 3H*, *Figure 3—figure supplement 2E*).

Due to SoxF redundancy in arteriovenous specification, an A/V malformation phenotype is typically only observed in double loss of Sox7 and Sox18 function. Since **Sm4** appeared to partially interfere with Sox7-Rbpjand Sox7-Sox18 PPIs in vitro, we turned to a Sox7 specific phenotype to assess whether this TF activity was inhibited by **Sm4** in vivo. The hallmark of **sox7** genetic disruption is a short circulatory loop in the head formed by the lateral dorsal artery (*Mohammed et al., 2013*), resulting in perturbed facial circulation (*Hermkens et al., 2015*). In presence of **Sm4**, we observe minor malformation to the LDA reminiscent of a partial Sox7 loss of function phenotype (Author response image 1). However, the blood circulation in the head is unaffected in **Sm4**-treated larvae, signifying that a short circulatory loop has not fully formed. This phenotype supports of the conclusion that Sox7 activity is only partially affected in presence of the small compound. Overall, these results are congruent with the genome-wide inhibitory effects observed in vitro, demonstrating that **Sm4** selectively interfered with the transcriptional activity of Sox18 and SoxF-mediated vascular formation in vivo.

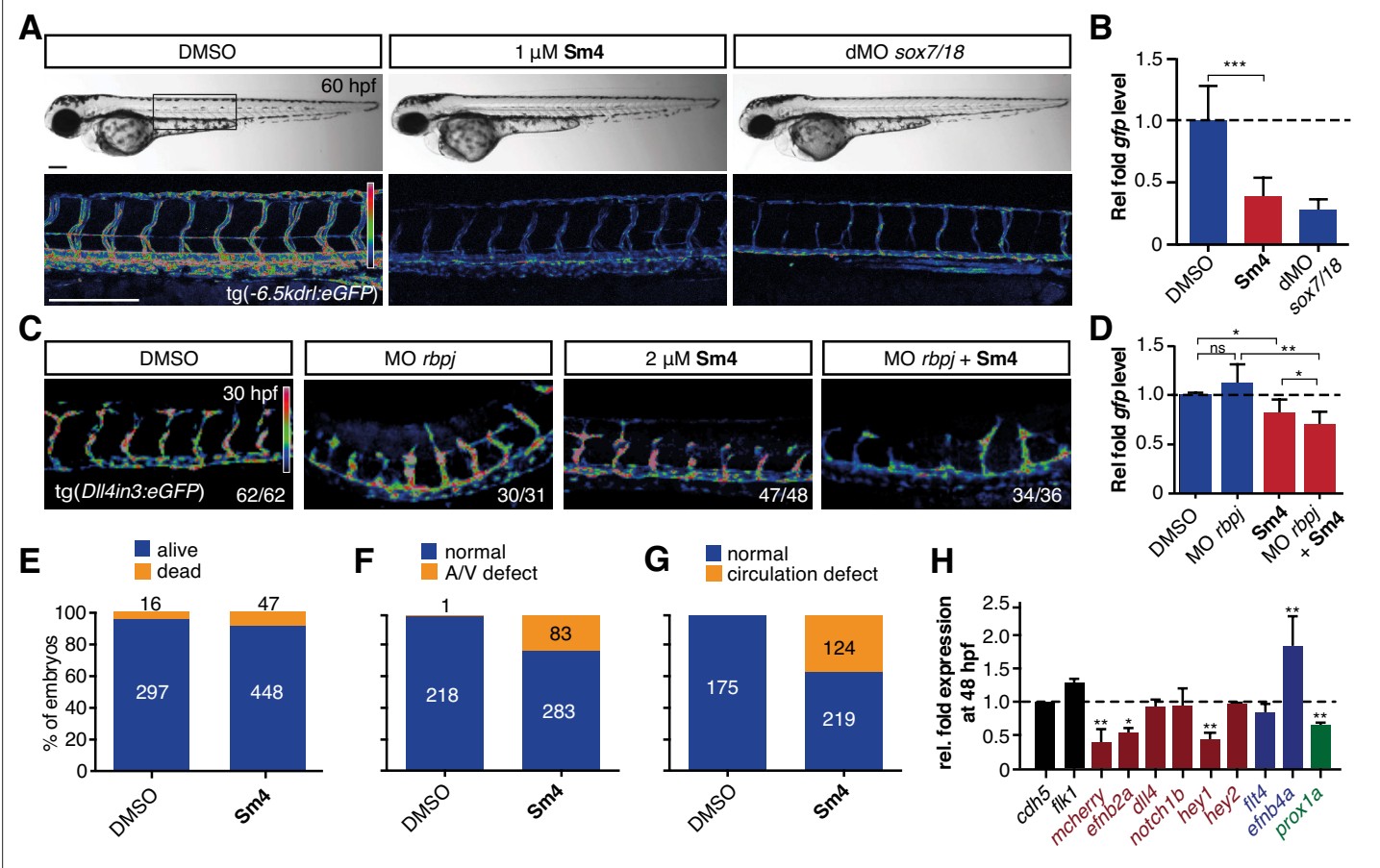

**Figure 3.** **Sm4** blocks SoxF transcriptional activity in vivo. (**A**) Lateral brightfield (top) and fluorescent (bottom) images of 60 hpf zebrafish larvae carrying the *tg(—6.5kdrl:eGFP)* SoxF reporter. Treatment was initiated at late stage (20 hpf) with either DMSO (negative control) or 1 mM **Sm4**, or larvae were injected with morpholinos against both *sox7* and *sox18* (dMO *sox7/18*). Fluorescence intensity is shown as heatmap. Scale bar 200 Mm (**B**) qRT-PCR analysis on *gfp* transcripts levels in treated *tg(—6.5kdrl:eGFP)* larvae and *sox7/18* morphants, showing reduction of activity on this transgene. (**C**) Lateral view of zebrafish larvae carrying the *tg(Dll4in3:eGFP)* SoxF/Notch reporter that harbors multiple binding sites for Rbpj and SoxF transcription factors. Larvae were injected with a morpholino against *rbpj* and/or treated with 2 mM **Sm4** from 13 hpf. (**D**) qRT-PCR analysis on *gfp* transcripts in *tg(Dll4in3: eGFP)* larvae, showing repression of combined SoxF/Notch activity in the **Sm4**-treated larvae. (**E**) Quantitation of embryonic lethality in larvae, treated with **Sm4** or DMSO control from early stage (16 hpf) until 72 hpf. (**F**) Penetrance of vascular phenotype (arteriovenous shunting) in 48 hpf larvae treated with 1.5 mM **Sm4** from 16 hpf. (**G**) Penetrance of circulation defect in 48 hpf larvae treated with 1.5 mM **Sm4** from 16 hpf. (**H**) qRT-PCR analysis of endogenous endothelial transcript levels at 48 hpf in larvae treated with 1.5 mM **Sm4** at 16 hpf, relative to DMSO control (dotted line). Data shown are mean ± s.e.m. *P<0.05, **P<0.01, ***P<0.001.

The online version of this article includes the following figure supplement(s) for figure 3:

**Figure supplement 1.** Sox9 activity is not perturbed by treatment in vivo.

**Figure supplement 2.** Sm4 interferes with SoxF activity in vivo.

As a final demonstration of the anti-angiogenic potential of **Sm4** in a therapeutically relevant setting, we next assessed its efficacy in a preclinical model of breast cancer. BALB/c mice were inoculated with highly metastatic 4T1.2 mammary carcinoma cells into the mammary fat pad, and three days were allowed for the engraftment of the tumor, after which treatment was initiated with either 25 mg/kg/day of **Sm4**, aspirin or vehicle PBS (*Figure 4A*). Aspirin was chosen as a negative control because of the structural similarity to **Sm4**. Daily treatment was maintained for a duration of 10 days, after which the primary tumor was resected and effects on disease latency were monitored (*Figure 4A*). As an indirect indication of target engagement, we first confirmed the expression of **Sox18** in the 4T1.2 tumor vasculature by in situ hybridization (*Figure 4B*). We next went on to measure **Sm4** bioavailability during the course of the treatment. **Sm4** was consistently detected in blood plasma at two

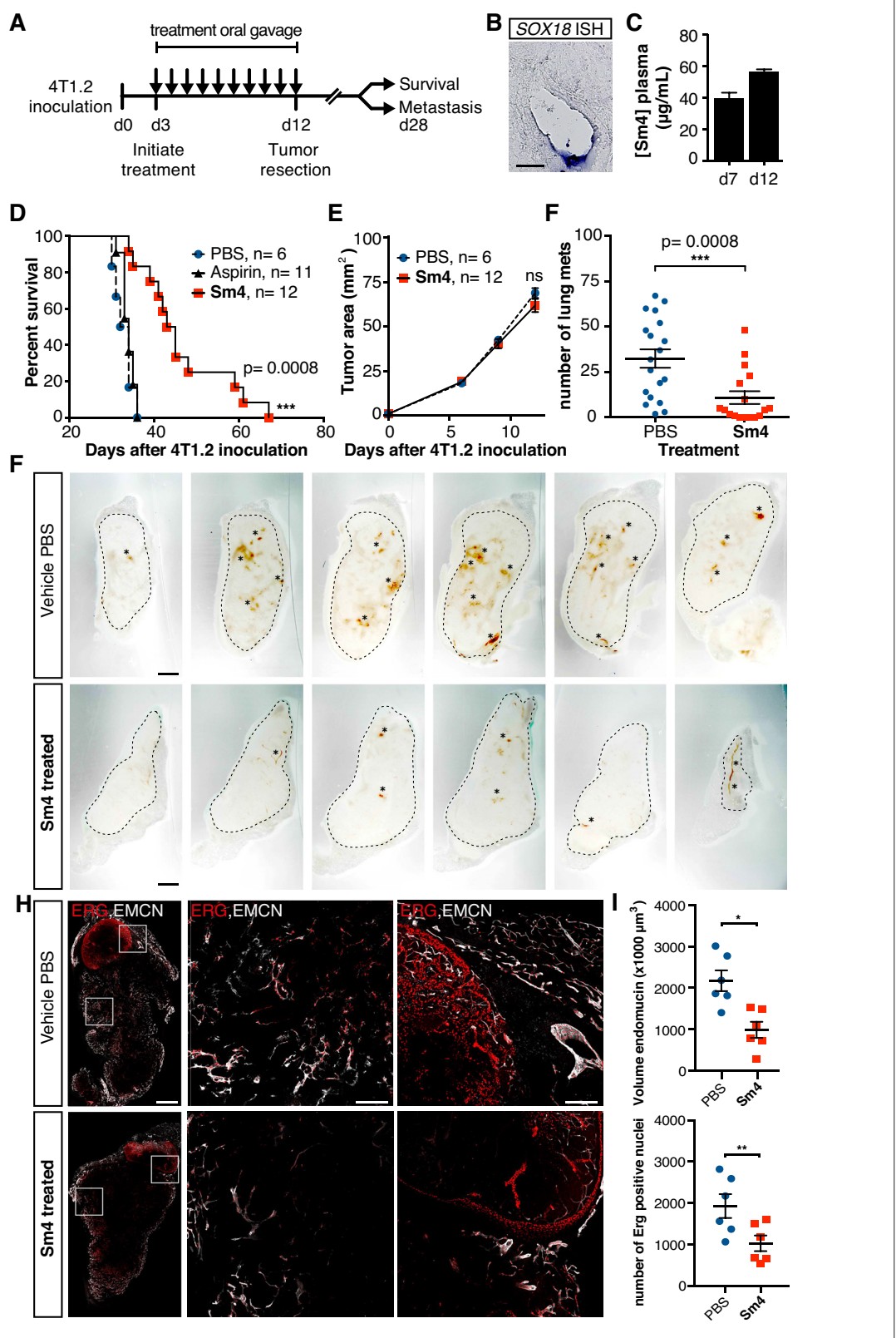

**Figure 4.** Metastasis and tumor vascularization is suppressed by **Sm4** treatment. (**A**) Timeline of mouse model for breast cancer metastasis. 4T1.2 tumor was inoculated at day 0, and resected at day 12. **Sm4** (25 mg/kg/day), Aspirin (25 mg/kg/day) or vehicle control (PBS), was administered orally on a daily basis from day 3 to day 12. Independent experiments were carried out to assess survival and metastatic rate. (**B**) Blood plasma concentrations

*Figure 4 continued on next page*

*Figure 4 continued*

of Sm4 during the course of the treatment scheme (day 7 and day 12) indicate good systemic delivery of the drug. (**C**) Expression of *SOX18* in the vasculature of the tumor as shown by in situ hybridization. Scale bar 100 Mm. (**D**) Survival of the mice was monitored (n=6–12 mice per group). Improved survival in Sm4-treated mice over both vehicle control and aspirin was analysed by Log-rank test ($P<0.001$). (**E**) No significant differences were found in tumor size at any stage. (**F**) Metastatic tumor nodules on the surface of the lungs were quantified at day 28, before any of the vehicle control or **Sm4**-treated animal had succumbed to the cancer burden. Data shown are mean ± s.e.m of 12–14 mice per group. (**G**) Vascular density was investigated on 300 Mm sections of whole tumors. Bright field images show the overall morphology of the tumor (outlined by dashed line) and presence of red blood cells, marking the main blood vessels and haemorrhagic areas (asterisks). Scale bar 1 mm. (**H**) Double immunofluorescence staining for endothelial specific markers ERG and Endomucin (EMCN) reveals the vascular patterning and penetration in the intra- and peri- tumoral regions. Left: whole tumor section (scale bar 1 mm), middle and right: blow-up of boxed regions (scale bar 200 Mm). (**I**) Quantitation of EMCN volume (blood vessel density) and ERGpositive nuclei (number of endothelial cells) of n=6 tumours per condition. Each data point represents the average of 3–4 representative regions (boxed areas in panel H) per tumor. Mean ± s.e.m for both conditions are shown. *$P<0.05$, **$P<0.01$.

The online version of this article includes the following figure supplement(s) for figure 4:

**Figure supplement 1.** Penetrance of blood vessels into 4T1.2 tumors is impaired by **Sm4**.

**Figure supplement 2.** Sm4-treated mice have decreased tumor vascular density.

**Figure supplement 3.** Sm4 treatment disrupts tumour-induced lymphangiogenesis.

different time points, with a mean concentration increasing over time from 38.3 Mg/ mL to 55.2 Mg/ mL (*Figure 4C*).

PBS vehicle- or aspirin-treated mice succumbed to the 4T1.2 tumor burden with a median latency of 33 and 34 days respectively (*Figure 4D*), whereas **Sm4**-treated mice had a significant increase in their overall survival with a median latency of 44 days (p-value <0.01). To further investigate what could cause such an effect, the size of the tumors was monitored during the treatment, as well as the formation of spontaneous lung metastases. While the size of the primary tumor was unchanged by **Sm4** treatment (*Figure 4E*), we found a 67% reduction in the mean number of lung metastases at day 28 after tumor inoculation (*Figure 4F*).

The lack of inhibition of **Sm4** on the primary tumor growth (*Figure 4E*) suggests that a potential combined effect between the drug treatment and surgery-induced inflammation is unlikely to be responsible for the increased survival, given that surgery is required on day 0 to inoculate the xeno-graft cancer cells into the mammary fat pad.

In order to establish a correlation between the metastatic rate and a tumor induced vascular response, we investigated the blood vessel density in the intra-tumoral and peri-tumoral regions (*Figure 4G*, *Figure 4—figure supplement 1*). Whole tumors were sectioned, and brightfield micros-copy revealed an overall reduction in blood vessel coverage, as indicated by the presence of red blood cells (*Figure 4G*, asterisks). Further analysis using immunofluorescent staining for endothelial cell markers ERG (nuclear) and Endomucin (EMCN, membranous), showed a significant decrease in the number of endothelial cells (48%, p-value <0.05), as well as the volume of the blood vessels (55%, p-value <0.01) in the tumors of **Sm4**-treated mice (*Figure 4H,I*, *Figure 4—figure supplement 2*). Using lymphatic specific markers PROX1 and podoplanin (PDPN), we also assessed the effect of **Sm4** on the tumor induced lymphangiogenic response, and found that the density of the tumor associated lymphatic vessels was greatly reduced (65%, p-value <0.01) in treated conditions, as well as the number of lymphatic endothelial cells (70%, p-value <0.001) (*Figure 4—figure supplement 3*). This lymphatic response to **Sm4**-treatment is consistent with that of SOX18 loss of function during lymphatic spread of solid cancers (*Duong et al., 2012*) Together, this demonstrates that **Sm4** improved the outcome of induced breast cancer by interfering with tumor-induced neo-vascularization and associated metastasis.

Induction of angio- and lymphangiogenesis is a hallmark of solid cancer, and is a critical step towards enabling tumor metastatic dissemination. Conventional approaches to target transcription factors have focused on interfering with oncogenes that are dysregulated to promote tumor cell transformation (*Gormally et al., 2014*; *Illendula et al., 2015*; *Moellering et al., 2009*; *Zhang et al., 2012*). Here, we validate a novel complementary strategy that relies on targeting a developmental

transcription factor from the host vasculature that can facilitate metastatic spread. Our results provide a proof of concept that targeting the transcription factor SOX18 with **Sm4** is an effective molecular strategy to interfere with the metastatic spread in a pre-clinical model of breast cancer.

## Materials and methods
### Experimental reproducibility
All data and statistical analysis in this study were generated from at least three independent experiments unless indicated otherwise. Technical replicates were included in every experiment to reduce background noise and detect technical anomalies. Samples of distinct experimental conditions were not exposed to any specific method of randomization, and groups were assessed under non-blinded conditions.

### Plasmid preparation for cell-free expression
The genetically encoded tags used here are enhanced GFP (GFP), mCherry (Cherry) and cMyc (myc). The proteins were cloned into the following cell free expression Gateway destination vectors respectively: N-terminal GFP tagged (pCellFree_G03), N-terminal Cherry-cMyc (pCellFree_G07) and C-terminal Cherry-cMyc tagged (pCellFree_G08) (*Gagoski et al., 2015*).The Open Reading Frames (ORFs) corresponding to the human SOX7 (BC071947), SOX17, RBPJ (BC020780) and MEF2C (BC026341) were sourced from the Human ORFeome collection version 1.1 and 5.1 or the Human Orfeome collaboration OCAA collection (Open Biosystems) as previously described and cloned at the ARVEC facility, UQ Diamantina Institute. The entry clones pDONOR223 or pENTR201 vectors were exchanged with the ccdB gene in the expression plasmid by LR recombination (Life Technologies, Australia). The full-length human **SOX18** gene was synthesized (IDT) and the transfers to vectors was realized using Gateway PCR cloning.

### Cell-free protein expression
The translation competent *Leishmania tarentolae* extract (LTE) was prepared as previously described (*Mureev et al., 2009*; *Kovtun et al., 2011*). Protein pairs were co-expressed by adding 30 nM of GFP template plasmid and 60 nM of Cherry template plasmid to LTE and incubating for 3 hr at 27 °C.

### ALPHA-Screen assay
The ALPHA-Screen Assay was performed as previously described (*Sierecki et al., 2014*), using the cMyc detection kit and Proxiplate-384 Plus plates (PerkinElmer). A serial dilution of each sample was measured. The LTE lysate co-expressing the proteins of interest was diluted in buffer A (25 mM HEPES, 50 mM NaCl). For the assay, 12.5 ᴍL (0.4 Mg) of Anti-cMyc coated Acceptor Beads in buffer B (25 mM HEPES, 50 mM NaCl, 0.001% NP40, 0.001% casein) were aliquoted into each well. This was followed by the addition of 2 pL of diluted sample and 2 pL of biotin labeled GFP-Nanotrap in buffer A. The plate was incubated for 45 min at RT. Afterward, 2 pL (0.4 pg) of Streptavidin coated Donor Beads diluted in buffer A, were added, followed by incubation in the dark for 45 min at RT. The ALPHA-Screen signal was obtained on an Envision Multilabel Plate Reader (PerkinElmer), using the manufacturer's recommended settings (excitation: 680/30 nm for 0.18 s, emission: 570/100 nm after 37ms). The resulting bell-shaped curve is an indication of a positive interaction, while a flat line reflects a lack of interaction between the proteins. The measurement of each protein pair was repeated a minimum of three times using separate plates. The Binding Index was calculated as:

$$BI = \left[ \frac{I - I_{neg}}{I_{ref} - I_{neg}} \right] x\ 100$$

For each experiment, I is the highest signal level (top of the hook effect curve) and $I_{neg}$ is the lowest (background) signal level. The signals were normalized to the $I_{ref}$ signal obtained for the interaction of SOX18 with itself.

For PPI disruption assay, protein pairs expressed in LTE were incubated for 1 hr with 100 pM **Sm4** or DMSO alone (0.7% DMSO final). 100 pM **Sm4** or DMSO was also added to buffer B. PPI disruption was calculated as: $\left( 1 - \frac{I_{Sm4}}{I_{DMSO}} \right) x\ 100$.

For IC50 determination, the assay was identical but a dilution range of **Sm4** was used (0.3–300 pM). Percentage of interaction was calculated as: $\frac{I_{Sm4}}{I_{DMSO}} x$ 100. Data from at least three independent experiments were fitted in GraphPad Prism (RRID: SCR_007370) version 6.0 using 3-parameter nonlinear regression.

## Cell culture and transfection

COS-7 cells were purchased from ATCC (CRL-1651, RRID: CVCL_0224) cultured at 37 °C, 5% $CO_2$ in DMEM (Life technologies, 11995) with added FBS, sodium pyruvate, L-glutamine, penicillin, streptomycin, non-essential amino acids and HEPES (N-2-hydroxyethylpiperazine-N'—2-ethanesulfonic acid). COS-7 cells were transfected for 4–6 hr, and incubated for another 24 hr before lysis and luciferase assay (Perkin Elmer, 6016711). Human umbilical vein endothelial cells (HUVECs) were purchased from Lonza Australia (CC-2519A). HUVEC for ChIP-MS, ChIP-seq and RNA-seq analyses were transfection for 7 hr and incubated another 14 hr. During small molecule treatment, cells were grown in medium containing low serum (0.4% FBS). HUVECs were cultured at 37 °C, 5% $CO_2$ in EGM-2 media supplemented according to the EGM-2 bullet kit instruction (Lonza, CC-3162). Cells for were grown in 35 mm dishes to 80–90% confluency, and transfected with plasmid mouse pSG5 *Sox18*, plasmid pSG5 *cMyc-Sox18*, or plasmid *cMyc* using X-tremegene 9 DNA transfection reagent (Roche, 06365787001) according to the manufacturer's instructions. All cell lines were tested negative for mycoplasma contamination.

## Chromatin immunoprecipitation

ChIP experiments were performed as previously described (*Schmidt et al., 2009*). Immunoprecipitation was performed using Anti-cMyc (Cell Signaling, #2276, RRID: AB_2314825) on HUVECs overexpressing cMyc-tagged SOX18.

## ChIP-seq and analysis

Following IP, DNA amplification was performed using TruSeq ChIPseq kit (Illumina, IP-202–1012), using 0.5 pM of the universal reverse PCR primer and the forward PCR primer containing the index sequence of choice in 50 pL 1 x NEBNext High-Fidelity PCR Master Mix (New England Biolabs, M0541). The number of PCR cycles ranged from 13 to 18, depending on the ChIP efficiency. The PCR product was purified using AMPure beads (1.8 vol) and eluted in 20 pL of resuspension buffer (Tris-Acetate 10 mM pH 8). The library was quantified using the KAPA library quantification kit for Illumina sequencing platforms (KAPA Biosystems, KK4824) and 50 bp single end reads were sequenced on a HiSeq2500 following the manufacturer's protocol. Illumina fastq files were mapped to the GRCh37/ UCSC hg19 genome assembly using bowtie, and peaks were called using MACS version 2.1.0. using input. To avoid false positive peaks calling due to the cMyc epitope, ChIP-seq with the cMyc epitope only were performed in parallel to SOX18-cMyc ChIP-seq and peaks called in these experimental conditions were substracted to the peaks called in the SOX18-cMyc conditions.

Genomic Regions Enrichment of Annotations Tool (GREAT, RRID: SCR_005807) was used to analyse the functional significance of cis-regulatory regions. ChIP-seq data are available in the ArrayExpress database (https://www.ebi.ac.uk/arrayexpress, RRID: SCR_002964) under accession number E-MTAB-4480 (SOX7) and E-MTAB-4481 (SOX18).

## ChIP-MS (RIME)

ChIP-MS experiments were performed as previously described (*Mohammed et al., 2013*). Peptides common between SOX18-cMyc and the negative control (cMyc-only) were binned and only peptides that were uniquely detected in the SOX18-cMyc transfected cell were considered for analysis.

## RNA-seq and analysis

Quadruplicate samples were processed for whole transcriptome sequencing using TruSeq stranded total RNA library prep kit (Illumina). Reads were mapped to the hg19 reference human genome using STAR aligner (*Dobin et al., 2013*), and only uniquely aligned reads were considered. Transcripts were assigned to genes using htseq_count (HTseq package) (*Anders et al., 2015*), and differential expression was calculated using DEseq2 (*Love et al., 2014*). Genes with adjusted p-value <0.05 were considered significant.

Differentially expressed genes were identified between **Sm4**-treated and DMSO control in SOX18 over-expressing cells, and separated in up-regulated and down-regulated (DOWN) genes. The locations of their transcription start sites (TSS) were correlated to the locations of transcription factors binding events that are available from the ENCODE consortium (RRID: SCR_006793), and from the SOX18 and SOX7 ChIP-seq experiment we performed in this study. To ensure that the TSSs were independent, a TSS was allowed to only be assigned to 1 ChIP-seq peak. Transcripts with >2 fold absolute fold change (log2FC >1 or < —1) were included for distance to TSS analysis. The median distance between the TSSs and binding events was compared to the expected distance of a set of randomly selected genes to obtain the median ratio. The control set of genes was selected from the pool of genes expressed in HUVECs so that they had a similar distribution of expression levels. To ensure that no bias was introduced by potential co-regulation of genes by SOX18 and any other transcription factor analysed, we subtracted genes with SOX18 peaks from the analyses for other transcription factors. The reverse analysis was also performed, subtracting genes containing c-JUN peaks from the analysis for SOX18. RNA-seq data are available in the ArrayExpress database (https://www.ebi.ac.uk/biostudies/arrayexpress) under accession number E-MTAB-4511.

## Quantitative RT-PCR

Total RNA was extracted using RNeasy mini kit (Qiagen, 74106) according to the manufacturers protocol, including on column DNA digestion. cDNA was synthesised from 1 p,g of purified RNA using the high capacity cDNA reverse transcription kit (Life Technologies, 4368813). Amplification and quantitation of target cDNA was performed in technical triplicate of at least three biological replicates using the SYBR green (Life Technologies, 4312704) methods. Reactions were run in 10 pL in 384-well plates using the ViiA 7 Real-Time PCR system. Housekeeper genes (*fi-actin* for *tg(Dll4in3: eGFP)*, ef1a for tg(—6.5kdrl:eGFP), chd5 for tg(fli1a:eGFP,—6.5kdrl:mCherry), *RPL13* and *GAPDH for HUVECs*) were selected based on the stability of their expression throughout the set of experimental conditions, or chosen on grounds of their vascular expression to normalize to endothelial cell content. Primer efficiencies were calculated using LinRegPCR, and amplification data was analysed using ViiA7 software and the Q-gene PCR analysis template.

## Zebrafish aquaculture and analysis

Zebrafish were maintained as previously described (*Hogan et al., 2009*), and all procedures involving animals conformed to guidelines of the animal ethics committee at the University of Queensland (IMB/030/16/NHMRC/ARC/HF) or were approved by local ethical review and licensed by the UK Home Office (PPL 30/2783 and PPL 30/3324). The *tg(—6.5kdrl:eGFP), tg(fli1a:eGFP,—6.5kdrl: mCherry)* and *tg(Dll4in3:GFP)* were previously described (*Sacilotto et al., 2013*; *Duong et al., 2014*; *Lawson and Weinstein, 2002*).

Dechorionation was performed by treatment with 25 pg/mL or 5 pg/mL pronase for 2 hr, or overnight, respectively. Zebrafish larvae were anesthetized using 0.01% tricaine. Representative larvae were embedded in 0.5% low-melting point agarose and imaged with the Zeiss LSM 710 confocal microscope.

## Zebrafish in situ hybridization and sectional analysis

Wholemount zebrafish (28 and 48 hpf) in situ hybridization was performed as previously described (*Thisse and Thisse, 2008*) with probe templates for *dab* (*Song et al., 2004*) and *ephrinB2a* (*Durbin et al., 1998*). Yolk sac was removed prior to addition of in 70% glycerol. For transverse sections, whole larvae where embedded in 4% agarose, sectioned at 150 pm using the Leica VT1000 S vibrating microtome. Imaging was performed on the Olympus BX-51 brightfield microscope (ISH), and Zeiss LSM 510 confocal microscope. For fluorescent images, larvae were DAPI-stained before embedding.

## Small molecule treatment and morpholino injections

All treatment with putative small molecule inhibitors, and corresponding control conditions, were performed in the presence of low concentration of DMSO (<1% v/v) to achieve reliable homogeneous solutions, and were prepared from 10 mM DMSO stock. For cell culture, small molecules were added to fresh media directly following transfection and cells were grown in this media until time-point of cell harvesting. For in vivo experiments involving zebrafish, compound treatment was initiated at the

designated timepoints by replacing the media, and media +compound was refreshed daily for the duration of the experiment. PTU treatment (0.003%) was done in parallel with the small molecules to block pigment formation when necessary. Previously published and validated morpholino oligomers against *sox7, sox18* (*Herpers et al., 2008*) and *rbpj* (*Sacilotto et al., 2013*) were micro-injected into single cell zebrafish zygotes at 5 ng for experiments performed with *tg (6.5kdrl:eGFP)* and *tg(fli1a:eG-FP,—6.5kdrl:mCherry)*, and 0.125–0.15 pmol suboptimal concentrations for experiments performed with *tg(Dll4in3:eGFP)*.

## Mice and mouse model

BALB/c wild-type (WT) were purchased from Walter and Eliza Hall Institute for Medical Research and used between the ages of 6 and 10 weeks. Mouse 4T1.2 mammary carcinoma cells were cultured in complete RPMI with 10% FBS in a 5% $CO_2$ incubator. $5x10^4$ 4T1.2 tumor cells were inoculated into the fourth mammary fat-pad of BALB/c WT mice as previously described (*Mittal et al., 2014*). Briefly, on day three after tumor implantation, mice were orally gavaged daily for 10 days with 25 mg/kg of body weight **Sm4**, aspirin or vehicle PBS. Tumor size was measured with a digital caliper as the product of two perpendicular diameters. Blood plasma was collected from mice on day 7 and 12, and **Sm4** concentrations were analyzed using a 4000 Qtrap LC-MS/MS system mass spectrometer. On day 12, mice were anesthetised to surgically remove primary tumor, or mice were put through surgery procedure with no excision of the primary tumor, and the wound was closed with surgical clips. Tumors were collected in formalin for histology. Lungs were harvested on day 28 and fixed in Bouin's solution for 24 hr and metastatic tumor nodules were counted under a dissection microscope. Survival of the mice was monitored in experiments where the lungs were not harvested. Groups of 6–14 mice per experiment were used for experimental tumor assays, to ensure adequate power to detect biological differences. All experiments were approved by the QIMR Berghofer Medical Research Institute Animal Ethics Committee (P1505).

For quantitation of the vasculature in the tumors, fixed tissues were embedded in 4% agarose and sectioned all the way through at 300 pm on a Leica VT1000 S vibrating microtome. Sections were collected on glass slides and imaged for bright field analysis on the penetration of perfused vessels. Subsequently, immunofluorescent staining was performed on sections using anti-mouse Endomucin (cat# sc-53941, RRID: AB_2100038), ERG (cat# ab92513, RRID: AB_2630401), PROX1 (AngioBio cat#11–002, RRID: AB_10013720) and Podoplanin (AngioBio cat#11–033, AB_2631191) antibodies. Whole tumor sections were imaged by acquiring a series of images along the z-axis using a 10 x objective on a Zeiss LSM 710 confocal microscope. Subsequently, high-resolution images were captured using a 20 x objective on 3–4 separate regions from each tumor, to account for heterogeneity of the vascular density within the tumors and minimise bias. Raw image files with identical dimensions (1274.87 pm x 1274.87 pm x 89.05 pm) were loaded into Imaris (Bitplane, RRID: SCR_007370), and processed using 'spots' function to count ERG or PROX1- positive nuclei and 'surface' to calculate volume or area of Endomucin or Podoplanin positive vessels. For each tumor (n=6), counts from the multiple regions were averaged and the data was plotted in Graphpad Prism 6.

## Acknowledgements

Professor Mark Smyth provided technical comments on the manuscript. MF is supported by a National Health and Medical Research Council of Australia (NHMRC) Career Development Fellowship (APP1111169). PK is supported by a NHMRC Senior Principal Research Fellowship (APP1059006). MAC is supported by an Australia Fellowship (AF51105). This work was supported by project grants from the NHMRC (APP1025082) to YG, (APP1048242 and APP1107643) to MF and the Cancer Council Queensland (1008392, 1048237 and 1048237) to MF, the Australian Research Council to MF (DP100140485), and to YG (FT110100478, DP130102396 and DP120101423) and C4D (IMB, The University of Queensland) to MAC. KH and JSC acknowledge the support of the University of Cambridge, Cancer Research UK and Hutchison Whampoa Limited. JSC is supported by an ERC starting grant. BLB is supported by grants HL064658 and HL089707 from the US NIH. RJ is supported by a 2013 MOST China-EU Science and Technology Cooperation Program (grant number 2013DFE33080), by the National Natural Science Foundation of China (grant number 31471238) and a 100 talent award of the Chinese Academy of Sciences. D M was supported by a Susan G Komen Breast Cancer Foundation Program Grant (IIR12221504). Confocal microscopy was performed at the

Australian Cancer Research Foundation Dynamic Imaging Centre for Cancer Biology. We thank Dr CL Hammond for kindly providing the col2a1:YFP transgenic reporter fish line. We thank Dr G Baillie (IMB, core sequencing facility) for providing technical support with RNA-seq analysis.

## Additional information

### Funding

| Funder | Grant reference number | Author |
| --- | --- | --- |
| University of Queensland | RHD student fellowship | Jeroen Overman |
| Susan G. Komen | IIR12221504 | Deepak Mittal |
| National Institutes of Health | HL064658 | Brian L Black |
| National Institutes of Health | HL089707 | Brian L Black |
| National Health and Medical Research Council | APP1059006 | Peter Koopman |
| National Natural Science Foundation of China | 31471238 | Ralf Jauch |
| National Health and Medical Research Council | APP1025082 | Yann Gambin |
| Australian Research Council | FT110100478 | Yann Gambin |
| Australian Research Council | DP130102396 | Yann Gambin |
| Australian Research Council | DP120101423 | Yann Gambin |
| National Health and Medical Research Council | APP1011242 | Mathias Francois |
| National Health and Medical Research Council | APP1048242 | Mathias Francois |
| Australian Research Council | DP140100485 | Mathias Francois |
| National Health and Medical Research Council | APP1111169 | Mathias Francois |
| Cancer Council Queensland | 1008392 | Mathias Francois |
| Cancer Council Queensland | 1048237 | Mathias Francois |
| Australian Research Council | DP100140485 | Mathias Francois |

The funders had no role in study design, data collection and interpretation, or the decision to submit the work for publication.

### Author contributions

Jeroen Overman, Mathias Francois, Conception and design, Acquisition of data, Analysis and interpretation of data, Drafting or revising the article, Contributed unpublished essential data or reagents; Frank Fontaine, Conception and design, Acquisition of data, Analysis and interpretation of data, Contributed unpublished essential data or reagents; Mehdi Moustaqil, Deepak Mittal, Natalia Sacilotto, Johannes Zuegg, Avril AB Robertson, Kelly Holmes, Angela A Salim, Sreeman Mamidyala, Mark S Butler, Ashley S Robinson, Acquisition of data, Analysis and interpretation of data; Emma Sierecki, Wayne Johnston, Kirill Alexandrov, Brian L Black, Sarah De Val, Robert J Capon, Jason S Carroll, Yann

Gambin, Acquisition of data, Analysis and interpretation of data, Contributed unpublished essential data or reagents; Emmanuelle Lesieur, Acquisition of data, Contributed unpublished essential data or reagents; Benjamin M Hogan, Drafting or revising the article, Contributed unpublished essential data or reagents; Timothy L Bailey, Conception and design, Contributed unpublished essential data or reagents; Peter Koopman, Conception and design, Drafting or revising the article, Contributed unpublished essential data or reagents; Ralf Jauch, Analysis and interpretation of data, Drafting or revising the article, Contributed unpublished essential data or reagents; Matthew A Cooper, Oversaw Med Chem part of the study - major intellectual input, Conception and design, Analysis and interpretation of data, Contributed unpublished essential data or reagents

**Author ORCIDs**
Brian L Black [ORCID] https://orcid.org/0000-0002-6664-8913
Jason S Carroll [ORCID] https://orcid.org/0000-0003-3643-0080
Yann Gambin [ORCID] https://orcid.org/0000-0001-7378-8976
Mathias Francois [ORCID] https://orcid.org/0000-0002-9846-6882

**Ethics**
All procedures involving animals conformed to guidelines of the animal ethics committee at the University of Queensland (IMB/030/16/NHMRC/ARC/HF) or were approved by local ethical review and licensed by the UK Home Office (PPL 30/2783 and PPL 30/451 3324).

**Decision letter and Author response**
Decision letter https://doi.org/10.7554/eLife.21221.SA1
Author response https://doi.org/10.7554/eLife.21221.SA2

## Additional files

**Supplementary files**
• Supplementary file 1. (A) GO term analysis (PANTHER) on top 5 K SOX18 ChIP-seq peaks, reveals over-representation of biological processes, which are in agreement with known roles for SOX18 (e.g. blood vessel morphogenesis, angiogenesis, blood vessel development). (B) Summary of sequencing statistics, listing the sample with the number of the replicate (#n). Percentage of mapped reads is consistently high across all samples (>87%). Mapping was performed with STAR aligner (*Dobin et al., 2013*). (C) Summary of endothelial specific TF expression levels and summary of distance from peak to TSS analysis on DE SOX18oe vs. **Sm4** genes. A subtraction of SOX18, or cJUN peaks from all TF peaks was performed to reduce overlap bias (column #2 and #3). **Sm4** down regulated genes are significantly closer to SOX18 and c-JUN ChIP-seq peaks.

**Data availability**
RNA-seq and ChIP-seq data are available in the ArrayExpress database.

The following datasets were generated:

| Author(s) | Year | Dataset title | Dataset URL | Database and Identifier |
|-----------|------|---------------|-------------|-------------------------|
| Overman J | 2016 | SOX7 ChIP-seq | https://www.ebi.ac.uk/arrayexpress/experiments/E-MTAB-4480 | ArrayExpress, E-MTAB-4480 |
| Overman J | 2016 | SOX18 ChIP-seq | https://www.ebi.ac.uk/arrayexpress/experiments/E-MTAB-4481 | ArrayExpress, E-MTAB-4481 |
| Overman J | 2016 | Sm4 RNA-seq | https://www.ebi.ac.uk/arrayexpress/experiments/E-MTAB-4511/ | ArrayExpress, E-MTAB-4511 |

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
