## [Decision Letter]

Thank you for submitting your article "Pharmacological targeting of the transcription factor SOX18 delays breast cancer in mice" for consideration by *eLife*. Your article has been favorably evaluated by Sean Morrison (Senior Editor) and three reviewers, one of whom, Holger Gerhardt (Reviewer #1), is a member of our Board of Reviewing Editors. The following individual involved in review of your submission has agreed to reveal their identity: Gou Young Koh (Reviewer #3).

The reviewers have discussed the reviews with one another and the Reviewing Editor has drafted this decision to help you prepare a revised submission.

General Assessment:

The reviewers find your study provides interesting new insights into protein-protein interactions of Sox family member transcription factors and shows convincingly that such interactions can be targeted with small molecules to interfere with good selectivity with transcription in vitro and in vivo. The demonstration of in vivo efficacy of the small molecular inhibitor in both a zebrafish model using transcriptional reporters and in a mouse tumour model in which you find significantly reduced metastasis are particularly remarkable. Together with the comprehensive identification and validation of protein-protein interactions, these results provide a significant advance that the reviewers believe will be of interest to a wider scientific audience.

Major Conclusions:

1) Despite their similarities Sox family member transcription factors engage in distinct protein-protein interactions.

2) Sox18 interacts directly with RBPjk and a number of other endothelial transcription factors in endothelial cells to regulate endothelial targets.

3) Small molecule SM4 (identified in a screen reported elsewhere) shows good efficacy and selectivity to inhibit a subset of the identified Sox18 protein-protein interaction and suppresses target transcription in vivo.

4) Targeting Sox18 PPI and thus its transcriptional activity is effective in vivo to reduce tumour angiogenesis and metastasis.

Whilst the reviewers find your results supports these major conclusions, they find that the following points should be adequately addressed before publication:

1) The claim of selectivity of SM4 against Sox18 deserves further investigation, in particular, to clarify if the potentially redundant activity of Sox17 in vascular development is also affected. Also, potential activity against Sox7 should be clarified. You may already have the required data on activity of SM4 against Sox17 and Sox7, in which case adding these to the figure and text, as well as commenting on their implications for the claimed selectivity of SM4 will be sufficient. In case this means you need to repeat experiments, we suggest to limit yourself to any that can be achieved reasonably within 2 months.

2) The discrepancy between the effects on the *Dll4int3* reporter and endogenous *Dll4* gene expression effects of SM4 need to be clarified. Given that the *Dll4int3* reporter does not contain all regulatory elements of the endogenous *Dll4* gene, it could be that SM4 is more effective against the reporter and other elements still drive endogenous *Dll4*. As you currently don't comment on the discrepancy in the text, it is difficult for us to recommend precisely how you should address this issue. We suggest to carefully compare the response of endogenous *Dll4* expression similarly to the dose response you find for the *Dll4* reporter. Given the dynamic nature of *Dll4* expression, it could also be a timing issue, and thus related to the half-life of the drug? A possible experiment in vivo could be to show in situ hybridisation for *Dll4* in WT fish treated with SM4. As *Dll4* levels have a major impact on blood vessel formation also in tumour angiogenesis, and your in vivo mouse experiments show effects on tumour angiogenesis, we feel this needs to be clear in order to understand the action of SM4. This should be possible to address in two months. In case experimental data do not provide a clear answer, please address this issue carefully in text and Discussion.

3) The mechanism of reduced tumour angiogenesis and metastasis should be adequately discussed and ideally experimentally clarified. Given that the route of metastasis is likely through lymphatics and that Sox18 also regulates lymphangiogenesis, we feel you should consider this as potential mechanism. As you have the expertise in mouse lymphatic analysis from previous studies, we hope you will be able to provide experiments that show whether or not SM4 interferes with tumour lymphangiogenesis through blocking Sox18 function. It will not be necessary to show that this is the definitive cause for changes in metastasis, but we feel that without analysing lymphatics, this work is incomplete. Given that you use a xenograft model, it should be feasible to perform these experiments within the 2 months of revision period.

We hope you will find these comments useful and clear to allow efficient revisions of the essential points. For your information, we also append the full set of reviews below.

*Reviewer #1:*

This interesting report by Overman et al. presents the functional characterisation of a pharmacological compound targeting the interaction of the transcription factor SOX18 with other proteins. This particular compound, Sm4, was isolated through a high-throughput screen for selective SOX18 inhibitors described fully in an accompanying manuscript that will be published elsewhere. The current manuscript used successive screening by first CHIP MS and then Α screen to map protein+protein interactions of Sox18, and then study in detail the selectivity and inhibitory capacity of the compound Sm4 on SOX18 activity both in vivo and in vitro. The in vivo used the zebrafish model in which Sox18 transcriptional regulation of known vascular genes was investigated in reporter lines and by QPCR. Finally, the authors used a mouse orthotopic mammary tumour model to test efficacy in tumour angiogenesis and metastasis in vivo.

Overall, the work convincingly demonstrates that the approach taken is valid to identify key protein protein interactions of transcription factors, and that targeting these interactions can be achieved with efficacy and some selectivity in vivo.

The work is well written and illustrated, and the major claims appear supported by the data. Having said that, there are a number of issues that should be addressed to further clarify the claim of the specificity of Sm4's inhibition of SOX18 direct protein interactions and its role in inhibiting Sox18-mediated vascular formation in vivo. Critically, the authors test specificity and efficacy of Sm4 in vivo first in the Dll4int3 reporter line, but endogenous Dll4 seems unaffected. This is left uncommented and raises questions regarding the mechanism at play.

Furthermore, whereas the arteriovenous development deficit is clearly demonstrated in fish, the rest of the vascular changes in fish and in mouse tumours don´t match with what would be expected if Sox18 and RBPJ activity was reduced in blood endothelium. Furthermore, whilst these vascular changes are well demonstrated in the tumours, the actual reason for reduced lung metastasis is not addressed. Surprisingly, despite having originally identified Sox18 as key regulator of lymphatic development, and given that metastatic spread from mammary tumours is largely driven by lymphatics, the authors did not look at and comment on tumour associated lymphatic changes and the potential effects of Sm4.

Given that the work primarily aims to establish the approach to target PPI for transcriptional regulation in vivo, the latter point may not necessarily have to be addressed experimentally, but should at least be discussed. More pertinent to the overall claims are the remaining questions on specificity.

Please find further detailed comments below:

1) Given the redundancy of Sox18 and Sox17 in the context of vascular development, the authors should include OCT4 interaction in their functional analysis of the effects of SM4 on both SOX17/OCT4 (reported interaction) and SOX18/OCT4.

2) – Could the authors describe what the arbitrary threshold used to define interaction in Figure 1C is? Were there any physiological or chemical reasons used in defining it?

3) – Could the authors comment about the significant proximity between c-FOS and EZH2 binding sites and the transcriptional start site of up-regulated genes following Sm4 treatment? Are their transcriptional activities significantly affected by Sm4 treatment?

4) The authors describe that "[…] we calculated the distance between the transcription start site (TSS) of a gene and a TF binding event, as a proxy for the likelihood of direct transcriptional regulation". But later mention that, in regard to this analysis, "The results indicate that Sm4 affected genes are dysregulated through a direct effect on SOX18 transcriptional activity". Given the 'proxy' nature of this experiment, I believe the claim could be turned down to better reflect the fact that this is a rather indirect indication of specificity.

5) Could the authors provide evidence that Sm4 does not affect SOX18 DNA-binding role? This particular piece of data is required to show that Sm4 role on Sox18 role in vivo indeed relies on affecting SOX18 interaction with other proteins rather that in its ability to bind to DNA.

6) Does Sm4 affect the transcriptional activity of Sox7? In Figure 3, could the authors please include a combined treatment with MO-sox18/Sm4 and MO-sox7/Sm4 to further demonstrate Sm4 specificity within SoxF factors. In addition, the authors should also include a MO-sox7/Mo-rbpj/Sm4 and MO-sox18/MO-rbpj/Sm4 controls in Figure 3C.

7) As mentioned in the overall assessment, does Sm4 treatment affect the pre-existing lymphatic vasculature in area surrounding the tumour? In addition, is neo-lymphangiogenesis affected?

*Reviewer #2:*

In this paper, the authors describe a specific inhibitor of the transcription factor SOX18, a known regulator of vascular development. Using a combination of in vitro/ChIP-seq/in vivo experiment, they build a nice scientific approach to develop an anti-angiogenic drug able to inhibit tumor development. They start from an elegant experiment of ChIP-MS to isolate nuclear factors interacting with SOX18 presumably on chromatin. They test also interaction of SOX18 with several transcription factors obtained from the ChIP-MS to validate the protein-protein interaction and also to measure the effect of a small molecule sm4, describe in another manuscript, on these interactions. A ChIP-seq experiment coupled to RNA-seq shows, using a good bio-informatics approach, the specificity at the genomic level of Sm4 on the SOX18 transcriptional activity. The authors also predict a possible relationship of SOX18 with C-Jun.

The second part of the manuscript, presents the in vivo validation of action of Sm4 on the genetic pathway controlled by SOX18 followed by experiments in mouse showing the inhibition of tumor vascularization by Sm4. To my knowledge, it is one of the first studies showing the development of a drug able to block specifically the activity of a SOX protein.

*Reviewer #3:*

Although the roles of SoxF members including Sox18 are crucial for formation and maintenance of vascular system, exactly how they form the protein complex for their transcriptional activities remains poorly elucidated. The present work may be the first report unveiling the entity of the proteins that physically interact with Sox18 and the interaction between SoxF members. Thus, this work provides a significant scientific advance by deciphering the mechanism of transcriptional regulation which are critical for vascular morphogenesis. Some additional in vivo data would improve the integrity of the manuscript for publication.

1) The authors demonstrated that a natural compound Sm4 effectively inhibits the transcriptional activity of Sox18 in molecular and cellular levels. Moreover, the inhibitory effect of Sm4 was verified in vivo by using zebrafish reporter lines (*tg(-6.5kdrl:eGFP)* and *tg(Dll4in3:eGFP)*), in which arteriovenous specification defects were assessed. All these zebrafish systems are known to be modulated by the cooperation of Sox7 and Sox18 rather than by Sox18 alone. Therefore, the authors should be cautious to claim the effect of Sm4 is dependent solely on Sox18 inhibition based on these complicated in vivo systems. This potential complexity should be discussed.

2) Sm4 appears to have a potential as an oral therapeutic agent to treat cancer as they have shown that Sm4 treatment reduced lung metastasis in a mouse orthotopic mammary tumor model. Reduced tumor angiogenesis is suggested as a working mechanism for the reduced tumor metastasis in this work. On the other hand, vascular destabilization is well-known to induce tumor metastasis. Additional analysis of tumor vessels integrity may provide clues as to how the vascular changes induced by Sm4 administration are associated with decreased metastasis.

3) The authors showed Sox18 expression in tumor vessels to rationalize the inhibitory effect of Sm4 on tumor angiogenesis. It has been previously reported that tumor angiogenesis/lymphangiogenesis was suppressed in Sox18-knockout mice (JNCI 2006); thus, pharmacological blockade of Sox18 can indeed reduce tumor angiogenesis. However, there is a possibility that Sm4 can inhibit tumor angiogenesis and tumor growth by affecting molecules other than Sox18. This possibility could be tested by administering Sm4 to Sox18-knockout mice bearing tumors, since the effect of Sm4 on Sox18 will be abolished in this mouse model.

---

## [Author Response]

[…] Whilst the reviewers find your results supports these major conclusions, they find that the following points should be adequately addressed before publication:(1) The claim of selectivity of SM4 against Sox18 deserves further investigation, in particular, to clarify if the potentially redundant activity of Sox17 in vascular development is also affected. Also, potential activity against Sox7 should be clarified. You may already have the required data on activity of SM4 against Sox17 and Sox7, in which case adding these to the figure and text, as well as commenting on their implications for the claimed selectivity of SM4 will be sufficient. In case this means you need to repeat experiments, we suggest to limit yourself to any that can be achieved reasonably within 2 months.

We acknowledge that selectivity is a crucial consideration for an inhibitor to be of use as a chemical probe or therapeutic. Therefore, we have broken down the matter regarding selectivity into three key aspects: PPI disruption, inhibition of transcriptional actvity, and SOXF loss of function phenocopy in vascular development.

PPI disruption selectivity

To further investigate the selectivity of Sm4 between all three SOXF family members (SOX7, -17 and -18) we have performed additional PPI analyses by ALPHAScreen, focusing on the recruitment of RBPJ (relevant in a vascular context as demonstrated by Sacilotto et al. 2013), MEF2C (only known SOX18 interactor, Hosking et al. 2001) and the SOX17 protein partner OCT4 (key determinant in endoderm specification, Jauch et al. 2011 and Aksoy et al. 2013).

Figure 1—figure supplement 1 (new): This in vitro protein-protein interaction analysis revealed that SOX18 has the capacity to selectively form a hetero-dimer with SOX7 and RBPJ whereas SOX17 does not interact with other SOXF proteins, nor does it interact with RBPJ or MEF2C. The interaction with RBPJ is conserved between SOX18 and SOX7 (left panel). Sm4 has the ability to interfere with SOX7-SOX18 heterodimer formation (IC_50_ 19.6 μM, Figure 1—figure supplement 1F) and partially disrupts SOX7-RBPJ interaction. In addition, as suggested by reviewer #1, we have investigated the effects of Sm4 on SOXF-OCT4 protein-protein interaction. Results show that Sm4 has the potential to disrupt SOX17-OCT4 interaction but not SOX18-OCT4 or SOX7-OCT4.

Data shown in Figure 1 of the manuscript combined with new supplemental results (new Figure 1—figure supplement 2) suggest that Sm4 selectivity is mostly towards SOX18 but has some potential to interfere with SOX7- or SOX17-dependent PPI (in the high micro-molar range [Sm4] 50-100 μM). Our observations indicate that this inhibitory effect predominantly occurs in the scenario where SOX18 and the other SOXF protein share a particular interactor.

Of note the interaction between SOX17 and OCT4 (POU5F1) is not relevant to endothelial cell biology since this transcription factor is not expressed by endothelial cells as shown by transcripts analysis from FANTOM5 database (Author response image 1), and from the RNA-seq data in HUVECs generated in-house (Figure 2). Correspondingly, OCT4 was not identified in SOX18 ChIP-MS experiments performed in endothelial cells. We can therefore exclude that the SOXF-OCT4 interaction – and disruption thereof – contributes to the observed effects of Sm4-treatment in endothelial cells.

**Author response image 1. sa2fig1:** Snapshot of FANTOM5 database, showing (absence of) OCT4 transcript levels in arterial, venous and lymphatic endothelial cell types.

It is often challenging for small molecules to achieve selective targeting between closely related proteins, such as those within the SOXF group of transcription factors (SOX7,-17 and 18). The small molecule that we describe acts in such a way that SOX18-dependent protein complexes are disrupted. Each SOXF protein has its distinct ‘primary’ function mediated by specific sets of PPIs, which bestows opportunities for selective inhibition. However, these 3 TFs also do have certain PPIs in common, which could explain their overlaping function in the context of rescue mechanism. Redundancy has been shown for SOX7, 17 and 18 protein that can act interchangeably to rescue the loss of function of one another within the F-group (Hosking et al. 2009). The fact that Sm4 has the potential to inhibit a subset of SOX7, SOX17 or SOX18–dependent PPIs is an advantage to prevent any potential redundancy mechanism.

The claims regarding the specific targeting of SOX18 by Sm4 has been reworded more carefully in light of these new data (fifth paragraph of the main text). The additional PPI analysis has been added as new Figure 1—figure supplement 2.

Off-target profiling

To include a wider analysis of Sm4 specificity towards SOX18, we have performed an unbiased off target investigation using a CEREP/Eurofins/ Panlabs profiling panel. This is shown by a new data set we have now included in the revised version of our companion manuscript (standard profiling panel, new Table S3 Fontaine et al. Cell Chemical Biology). Proteins tested on this panel are representative of various biological processes, such as: GPCRs, kinases, nuclear receptors, HDACs, sirtuins and membrane receptors. CEREP uses as a cut off 50% inhibition to flag any potential off target effect. This panel analysis with Sm4at 10 μM did not flag any non-specific binding out of 36 protein tested.

Selectivity on transcriptional interference

To assess other SOX proteins’ activity that could be potentially affected by Sm4 we have included *SOX9* and SOX17 as negative controls throughout the study. We show that Sm4 does not perturb:

*SOX9* homodimer formation (Figure 1)

*SOX9* transcriptional activity in cell-based assay in vitro (Figure 2—figure supplement 3)

*SOX9*-induced Col2a1 transactivation in zebrafish larvae (Figure 3—figure supplement 1)

SOX17-induced ECE1 transactivation (Figure 2—figure supplement 3).

The analysis of ChiPseq/RNAseq data sets (Figure 2) further demonstrates that Sm4 is specific to SOX18 interference amongst endothelial TFs, including SOX7.

Selectivity on vascular development- phenotypic output

The current data set based on two SoxF reporter assays in zebrafish (Figure 3), combined with the partial phenocopy of *sox7/18* double morphants/knockout (Figure 3—figure supplement 2) and the ALPHAScreen data, demonstrates the ability of Sm4 to block Sox18 activity in vivo. Since Sm4 has the ability to interfere partially with Sox7/RBPJ interaction, we further investigated whether Sm4 could directly interfere with Sox7 function in zebrafish. For this approach, we used as a readout the phenotypic outcome of the *sox7* KO zebrafish line (Hermkens et al. Development 2015) and compared it to Sm4-induced phenotype.

The hallmark of *sox7* genetic disruption in zebrafish is a short circulatory loop in the head with no circulation in the trunk and tortuous lateral dorsal artery (LDA). In presence of Sm4, we observe a partial phenocopy of Sox7 loss of function characterized by a mild vascular defect in the LDA (Author response image 2) . The observed Sm4-induced phenotype supports the conclusion that Sox7 activity is partially affected in presence of the small compound. However, the treated larvae fully establish blood circulation in the head (in contrast to the trunk), and do not form a short circulatory loop typical of Sox7 loss of function. This is now included in the text of the revised manuscript (eleventh paragraph of the main text). Of note, it is possible that the minor LDA phenotype is be secondary to the arteriovenous fusion phenotype.

**Author response image 2. sa2fig2:** Sm4-treatment causes mild malformations to the lateral dorsal aorta (LDA), reminiscent of partial interference with Sox7 function. Head circulation is unaffected by Sm4.

Lastly, the use of various zebrafish model system to assess the effects of Sm4 in vivo during development (Figure 3 and Figure 3—figure supplement 1 and Figure 3—figure supplement 2) strongly suggest that Sm4 has no conspicuous effect on other SOX TFs. Interference (chemical or genetic) with developmental transcription factors at the stages we investigated would results in severe defects, while we observe that Sm4-treated zebrafish larvae develop normally, with the exception of the phenotype associated with perturbed SOXF function (malformation of axial blood vessels).

2) The discrepancy between the effects on the Dll4int3 reporter and endogenous Dll4 gene expression effects of SM4 need to be clarified. Given that the Dll4int3 reporter does not contain all regulatory elements of the endogenous Dll4 gene, it could be that SM4 is more effective against the reporter and other elements still drive endogenous Dll4. As you currently don't comment on the discrepancy in the text, it is difficult for us to recommend precisely how you should address this issue. We suggest to carefully compare the response of endogenous Dll4 expression similarly to the dose response you find for the Dll4 reporter. Given the dynamic nature of Dll4 expression, it could also be a timing issue, and thus related to the half-life of the drug? A possible experiment in vivo could be to show in situ hybridisation for Dll4 in WT fish treated with SM4. As Dll4 levels have a major impact on blood vessel formation also in tumour angiogenesis, and your in vivo mouse experiments show effects on tumour angiogenesis, we feel this needs to be clear in order to understand the action of SM4. This should be possible to address in two months. In case experimental data do not provide a clear answer, please address this issue carefully in text and Discussion.

We agree with the interpretation regarding the discrepancy in terms of activity comparing *Dll4int3* synthetic reporter line versus endogenous *dll4* expression. It is often the case that synthetic enhancers – containing only a discrete number of regulatory elements – are more responsive to a subset of regulators. For example, the synthetic *-6.5kdrl* promoter that we use the assess SoxF activity is highly responsive to Sox7 and Sox18 activity, while the endogenous gene *kdrl (flk1*) is not (Duong et al. 2014). We also observe this for this synthetic promoter fragment using both Morpholino approaches and Sm4-treatment (Figure 3).

Work by Sacilotto et al. demonstrated that deletion of the Sox binding motif in this Dll4int3 enhancer fragment leads to a loss of transcriptional activation, which demonstrates that this transgene is dependent on SoxF activity (Sacilotto et al. 2013). This does not mean that the endogenous regulation of *dll4* solely relies on SoxF activity. It has been shown that another *dll4* enhancer *Dll4int12* is also required to drive proper expression of this gene in endothelial cells (Wyth et al. 2013). This enhancer relies on Ets and Rbpj combinatorial mode of action and it is unknown whether SoxF are at play to modulate this particular regulatory element.

In the context of vascular development in the zebrafish and mouse, Sacilotto et al. showed that individual loss of SoxF proteins or Rbpj has little effect on *Dll4in3* activation, and correspondingly, we show in our study that Sm4 treatment slightly affects this transgene in zebrafish (Figure 3). A more profound effect was observed when Sm4-treatment was combined with *rbpj* morpholino injections. Overall, we do not claim that Sox18 is a master regulator of *dll4* expression, nor do we suggest that Sm4 is a chemical regulator of *dll4* transcription. Instead, we utilize *Dll4int3* synthetic enhancer activity as a readout for on-target Sox18 inhibition. To make this clear to the reader, we have adjusted the text accordingly (tenth paragraph of the main text).

In order to explore the effect of **Sm4** on *dll4* endogenous transcriptional activation we have performed in situ hybridization on DMSO ctrl and Sm4 treated zebrafish larvae, as suggested (Author response image 3).

This analysis shows that the effect of Sm4 on the endogenous *dll4* transcript is not as profound as the effects observed on *Dll3int3* enhancer activity. This result is consistent with the qRT-PCR data analysis of *dll4* in Sm4-treated zebrafish larvae (Figure 4G), which also show a very mild reduction in the overall *dll4* transcript levels.

**Author response image 3. sa2fig3:** Effect of Sm4 on endogenous dll4 transcript in 27 hpf zebrafish larvae. Both the dorsal aorta and intersomitic vessels (ISV) were labeled by dll4 ish probe. In presence of Sm4(1 μM) ISV show a mild decrease of signal intensity.

3) The mechanism of reduced tumour angiogenesis and metastasis should be adequately discussed and ideally experimentally clarified. Given that the route of metastasis is likely through lymphatics and that Sox18 also regulates lymphangiogenesis, we feel you should consider this as potential mechanism. As you have the expertise in mouse lymphatic analysis from previous studies, we hope you will be able to provide experiments that show whether or not SM4 interferes with tumour lymphangiogenesis through blocking Sox18 function. It will not be necessary to show that this is the definitive cause for changes in metastasis, but we feel that without analysing lymphatics, this work is incomplete. Given that you use a xenograft model, it should be feasible to perform these experiments within the 2 months of revision period.

We greatly appreciate this suggestion, given the reported role of Sox18 in lymphangiogenesis and the contribution of lymphatic vasculature to the malignancy of many types of solid tumour. To address this request, we have quantified the tumour-induced lymphangiogenic response in absence or presence of Sm4. Immunofluorescence staining of 4T1.2 tumour sections was performed for lymphatic specific markers PROX1 and Podoplanin along with the blood vascular marker endomucin. Results show that both lymphatic vessel density and number of lymphatic endothelial cells is reduced in presence of Sm4 treatment. This lack of lymphatic outgrowth in presence of the small molecule is likely to contribute to the decrease in lung metastasis and improved disease latency. This result is now included in the manuscript as new Figure 4—figure supplement 3 and in the main text (fifteenth paragraph).